# Report

EMBO
reports

# Using a bistable animal opsin for switchable and scalable optogenetic inhibition of neurons

Jessica Rodgers[1] , Beatriz Bano-Otalora[1] , Mino D C Belle[2] , Sarika Paul[1], Rebecca Hughes[1], Phillip Wright[1], Richard McDowell[1], Nina Milosavljevic[1] , Patrycja Orlowska-Feuer[1,3] , Franck P Martial[1], Jonathan Wynne[1], Edward R Ballister[1,4] , Riccardo Storchi[1], Annette E Allen[1] , Timothy Brown[1] & Robert J Lucas[1,*]

## Abstract

There is no consensus on the best inhibitory optogenetic tool. Since Gi/o signalling is a native mechanism of neuronal inhibition, we asked whether Lamprey Parapinopsin ("Lamplight"), a Gi/o-coupled bistable animal opsin, could be used for optogenetic silencing. We show that short (405 nm) and long (525 nm) wavelength pulses repeatedly switch Lamplight between stable signalling active and inactive states, respectively, and that combining these wavelengths can be used to achieve intermediate levels of activity. These properties can be applied to produce switchable neuronal hyperpolarisation and suppression of spontaneous spike firing in the mouse hypothalamic suprachiasmatic nucleus. Expressing Lamplight in (predominantly) ON bipolar cells can photosensitise retinas following advanced photoreceptor degeneration, with 405 and 525 nm stimuli producing responses of opposite sign in the output neurons of the retina. We conclude that bistable animal opsins can co-opt endogenous signalling mechanisms to allow optogenetic inhibition that is scalable, sustained and reversible.

**Keywords** bistable; GPCR; neuronal inhibition; opsins; optogenetics
**Subject Categories** Methods & Resources; Neuroscience

## Introduction

Inhibitory optogenetic tools are light-sensitive proteins or opsins which can be expressed in neurons to cause hyperpolarisation and neuronal silencing (Wiegert *et al*, 2017; Ferenczi *et al*, 2019). There is currently no consensus regarding which tools are best for either research or therapeutic applications. Many have been based on light-sensitive ion channels and pumps. Early examples, such as the inwardly directed chloride pump Halorhodopsin (Han & Boyden, 2007; Zhang *et al*, 2007) and the outwardly directed proton pump Archaerhodopsin (Chow *et al*, 2010), provide hyperpolarisation with millisecond resolution, but require very bright light. Later generation tools, including the red-shifted chloride pump JAWS (Chuong *et al*, 2014) and anion conducting channelrhodopsins, such as gtACR1 and gtACR2 (Govorunova *et al*, 2015), are sensitive to lower levels of light, but still require repeated light stimulation to maintain neuronal inhibition. Step function tools, including the chloride-conducting channelrhodopsin SwiChR (Berndt *et al*, 2014) and the light-gated potassium channel Blink2 (Cosentino *et al*, 2015; Alberio *et al*, 2018), have been developed for sustained silencing over the second-minute timescale. However, there is still potential for unpredictable cellular responses using these tools, including non-physiological ion concentration gradients, antidromic spiking, (Mattis *et al*, 2011; Ferenczi & Deisseroth, 2012) and unexpected excitatory activity (Mahn *et al*, 2016; Malyshev *et al*, 2017). HyLighter (Janovjak *et al*, 2010) a potassium-selective light-gated ion channel also allows switchable inhibition that persists during dark, but requires high intensity UV light (380 nm) and supplementation with an azobenzene photoswitch in order to function.

A more physiological alternative would be to take advantage of endogenous cellular processes capable of causing neuronal inhibition, such as Gi/o signalling, which can activate G protein-coupled inwardly rectifying potassium (GIRK) channels to induce hyperpolarisation (Wettschureck & Offermanns, 2005; Armbruster *et al*, 2007). This widespread mechanism, downstream of native GPCRs providing inhibitory neuronal modulation, is co-opted by the inhibitory chemogenetic tool, hM4Di (Urban & Roth, 2015; Whissell *et al*, 2016). Gi/o signalling is also involved in GIRK-independent synaptic silencing (Lüscher *et al*, 1997; Stachniak *et al*, 2014).

Optogenetic control over Gi/o signalling could be provided by animal opsins, light-sensitive G protein-coupled receptors (GPCRs). Indeed, naturally occurring Gi/o-coupled opsins, such as rod opsin, cone opsins and the mu-Opioid receptor-rod opsin chimera,

1 Faculty of Biology Medicine and Health, University of Manchester, Manchester, UK
2 Institute of Biomedical and Clinical Sciences, University of Exeter Medical School, University of Exeter, Exeter, UK
3 Department of Neurophysiology and Chronobiology, Institute of Zoology and Biomedical Research, Jagiellonian University in Krakow, Krakow, Poland
4 Department of Biomedical Engineering, Columbia University, New York, NY, USA
*Corresponding author. Tel: +44 161 275 5251; E-mail: robert.lucas@manchester.ac.uk

opto-MOR, have already been used as optogenetic tools, mostly for restoring visual responses in the degenerate retina (Cehajic-Kapetanovic *et al*, 2015; Gaub *et al*, 2015; Wyk *et al*, 2015; Berry *et al*, 2019), which has also been accomplished using microbial opsins (Bi *et al*, 2006; Lagali *et al*, 2008; Sengupta *et al*, 2016), but also to produce neural inhibition in the brain (Airan *et al*, 2009; Masseck *et al*, 2014; Siuda *et al*, 2015). However, while these tools are activated by light, their deactivation relies on light-independent mechanisms making it difficult to maintain stable opsin activation and regulate timing of release from optogenetic inhibition. The natural diversity of animal opsins provides an attractive potential solution. "Bistable" animal opsins are thermally stable in both photoactivated and "dark" states and may be switched on and off by light, potentially allowing close control of the timing and degree of G protein signalling.

Here, we examine whether bistable animal opsins could form the basis of switchable inhibitory optogenetic tools. For this purpose, we used lamprey parapinopsin (Koyanagi *et al*, 2004, termed here Lamplight), which couples to Gi *in vitro* (Kawano-Yamashita *et al*, 2015) and forms active and inactive states with very different peak wavelength sensitivity, 370 and 515 nm, respectively (Wada *et al*, 2018; Eickelbeck *et al*, 2020). We show these characteristics mean Lamplight allows switchable and scalable inhibition of neurons using safer light intensities than alternatives based on ion channels or pumps (Perny *et al*, 2016).

## Results and Discussion

### Lamplight allows switchable, titratable control of Go activity

We first explored Lamplight's potential for switchable control over G protein signalling using a live cell bioluminescent resonance energy transfer (BRET) assay (Masuho *et al*, 2015) to report Go activation in Hek293T cells (Fig 1A). Lamplight expression in a subset of Hek293T cells was correctly localised to the plasma membrane (Fig 1A) and led to a clear increase in BRET signal in response to a 1 s 405 nm light (Fig 1B), confirming that Lamplight activates Go.

Animal opsins frequently drive responses at lower intensities than other optogenetic actuators, such as microbial opsins. BRET signal amplitude could be modulated across a range of 405 nm intensities, 13.5–15.5 log photons/cm$^2$/s, ~0.3–30 µW/mm$^2$ (Fig 1B). Lamplight sensitivity (log EC50 = 14.4 log photons/cm$^2$/s, 2.4 µW/mm$^2$, Fig 1C) was consistent with other animal opsins tested using the BRET assay (Rod opsin 480 nm log EC50 = 14.6 log photons, 3 µW/mm$^2$), around 1,000-fold less light than required by Halorhodopsin and Archaerhodopsin, 1–10 mW/mm$^2$ (Han & Boyden, 2007; Zhang *et al*, 2007; Chow *et al*, 2010). Signal amplification at later steps in the G protein signalling cascade could also render integrated cellular responses to Lamplight substantially more sensitive.

We then set out to determine how closely Go activity in this system could be controlled using Lamplight. The BRET signal in Lamplight-expressing cells remained elevated over tens of seconds following 1 s 405 nm illumination, suggesting continuous opsin signalling (as expected for a thermally stable active photoproduct). Presentation of a 1 s light pulse predicted to be optimally absorbed by Lamplight's activated state, 525 nm, resulted in relaxation of the BRET signal back to baseline (Fig 1D). Note that, although this relaxation occurred over several seconds, this time course likely reflects the time taken for uncoupling of nanoLuc-GRK3 fragment from the beta-gamma dimer, as the light-induced structural changes in the opsin that produce deactivation are expected to be rapid. Both previous (Eickelbeck *et al*, 2020) and our own experiments (below) demonstrate faster kinetics are possible depending on assay and cell type used. These characteristics suggest that Lamplight could function as a switchable optogenetic tool in this environment, and indeed a subsequent 405 nm pulse restored a sustained high BRET signal (Fig 1D).

We next sought to establish whether Lamplight could provide quantitative control over the level of Go activation. Having shown different intensities of 405 nm could be used to scale the amplitude of the Lamplight response (Fig 1B), we found a similar effect was possible starting from an activated state by modulating subsequent 525 nm exposure (Fig 1E). While a single 405 nm pulse established a high BRET signal that could be fully switched off with a subsequent bright 525 nm pulse, a lower intensity 525 nm pulse established an intermediate BRET signal that was sustained for the rest of the recording. Moreover, sequential exposures to the lower intensity 525 nm produced stepwise reductions in BRET signal.

---

**Figure 1.  Lamplight can be used for switchable and tuneable control of Go activity in Hek293T cells.**

A   Hek293T cells expressing Lamplight-1D4 (anti-1D4 = green, DAPI = blue, Top) and schematic of BRET assay, measuring interaction of liberated Gβγ-Venus with GRK3 fragment-Nanoluciferase (Bottom). Scale bar = 20 µm.

B   Lamplight-driven response to increasing intensity 1 s 405 nm light

C   405 nm irradiance response curve

D   Response to 1 s 405 nm (blue line) and 525 nm light (green line)

E   Response to 405 nm (blue line) followed by 525 nm light of different intensities (green line)

F   Schematic of Lamplight photoequilibrium. In dark, Lamplight is mostly in inactive (R) state. 405 nm light shifts equilibrium (blue arrow) so opsin is mostly in active (M) state, while 525 nm light shifts (green arrow) Lamplight into R state. Shifts caused by concurrent 525 and 405 nm partially cancel out leading to a small increase of opsin in M state.

G   Response to 405 nm alone or with 525 nm light

H, I   Representative time courses of Lamplight-driven responses to 8 s mixed stimuli (grey bar) with different 405 nm: 525 nm ratios from 1 (100% 405 nm) to 0 (100% 525 nm) from (H) dark or (I) after 8 s 405 nm light (blue bar)

J   Sustained Lamplight-driven activity (mean BRET ratio 20–80 s after stimulus) after exposure to mixed stimuli from dark (ON) or 405 nm light-adapted state (OFF). Lamplight data fit with linear trendline. Best fitting parameters for ON, slope = 0.311, Y-intercept = 0.04, $R^2$ = 0.85. For OFF, slope = 0.30, Y-intercept = 0.07, $R^2$ = 0.69.

Data information: For (B–E, G), error bars = SEM, N = 3–4 biological replicates. All intensities are for total Lamplight effective log photons/cm$^2$/s. Timing of light stimuli shown by vertical line/bar.

---

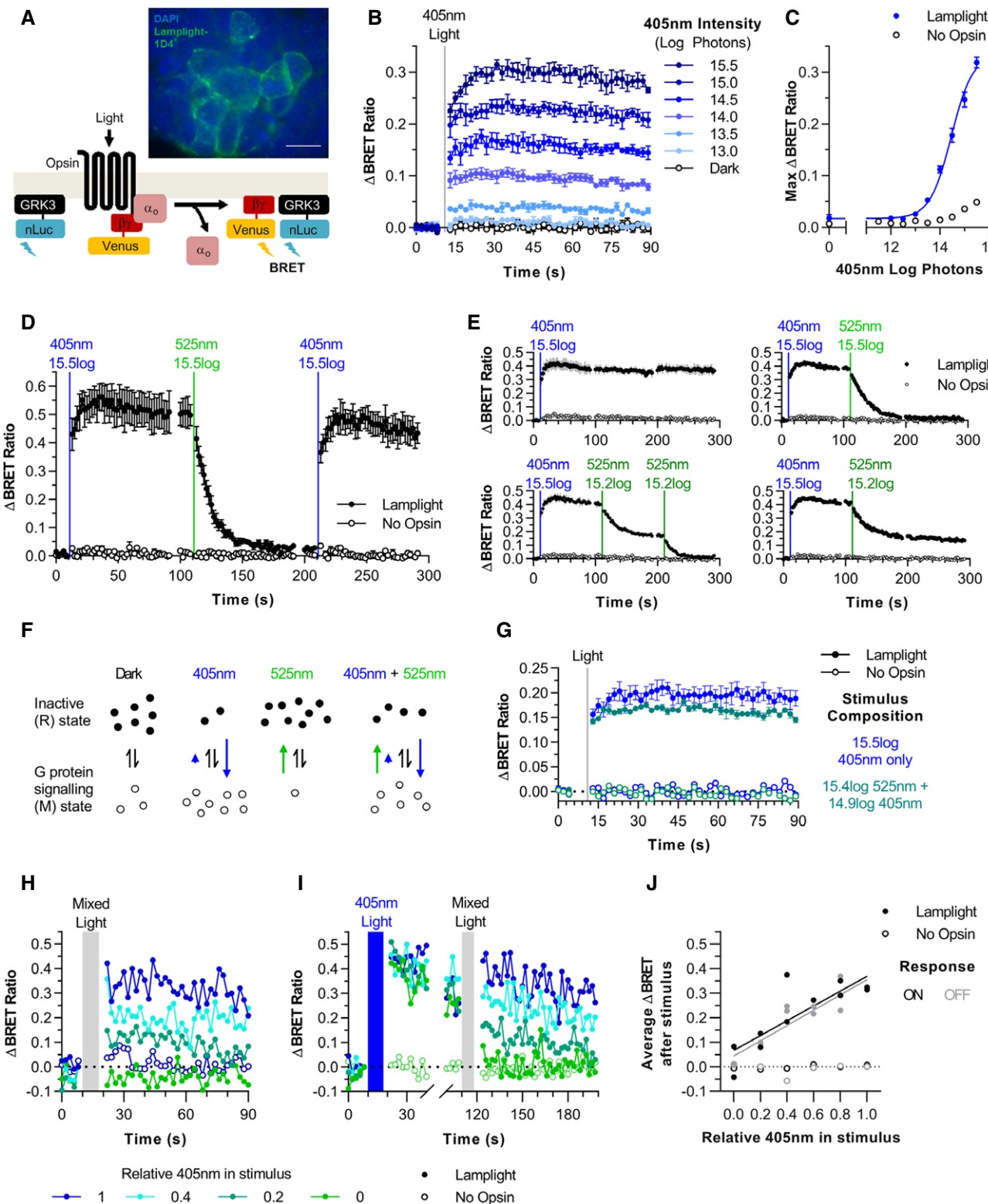

Figure 1.

An alternative approach to tuning signal activity for bistable opsins can be to adjust the ratio of activating versus deactivating wavelengths in incident light. Thus, under extended exposure, bistable opsins are expected to reach an equilibrium at which fractional concentrations of active and inactive states are defined by the spectral composition of the light. In the case of Lamplight, stimuli rich in short wavelengths should strongly bias the pool of available opsin towards the active state, while stimuli rich in long

wavelengths should enrich the inactive state (Fig 1F). In such a system, controlling the ratio of short to long-wavelength light can provide quantitative control over G protein signalling that is not dependent upon the starting state concentrations of the two opsin states (and thus prior light exposure).

Our first step in determining whether Lamplight showed such behaviour was to ascertain whether 525 nm light antagonised the effects of a 405 nm pulse in dark-adapted cells. This proved to be the case (Fig 1G). We next presented mixtures of 405 and 525 nm light at a range of ratios (see Materials and Methods), with stimulus duration increased to 8 s to maximise the possibility of reaching an equilibrium. We found that the BRET signal magnitude was impacted by the wavelength ratio, with larger responses elicited by stimuli biased in favour of 405 nm, in both dark-adapted cells and cells previously photoactivated by 405 nm light (Fig 1H and I). Indeed, these bispectral stimuli produced equivalent BRET signals for different initial conditions (Fig 1J). These data indicate that tuning the ratio of short versus long-wavelength light achieves quantitative control over Lamplight signalling independent of previous light exposure, which could be a useful complement to the standard "intensity" method, as it has the potential to provide a more uniform level of activity across a population of neurons differing in their amount of light exposure.

Our findings in Hek293T cells suggest that Lamplight allows switchable quantitative control over G protein signalling using two approaches: (i) changing the intensity of the activating or deactivating stimuli or (ii) varying the spectral composition of incident light stimulus.

## Lamplight as an inhibitory optogenetic tool for neuronal silencing

Having established Lamplight can be used for switchable control of Go activity in Hek293T cells, we turned to the question of whether this could be applied to neurons. We targeted a subset of Cre-expressing neurons in the hypothalamic suprachiasmatic nucleus (SCN) of $VIP^{Cre/+}$ mice with a floxed Lamplight-mCherry adeno-associated virus (AAV). Like other SCN neurons, these VIP-expressing cells spontaneously increase firing during the circadian light phase (Inouye & Kawamura, 1979; Green & Gillette, 1982; Meijer et al, 1996; Yamazaki et al, 1998; Hermanstyne et al, 2016), making them suitable for exploring inhibitory effects of Lamplight. Moreover,

these neurons are not readily silenced by the commonly applied optogenetic inhibitor, ArchT (Paul et al, 2020). At 4 weeks post-injection, we then made targeted whole-cell current-clamp recordings from fluorescent neurons in living SCN slices (Fig 2A).

Short wavelength light (405 nm) caused hyperpolarisation and suppression of spike firing in mCherry-labelled cells (Fig 2B), while non-fluorescent cells did not exhibit changes in resting membrane potential (RMP) or spike firing (Fig EV1A and B). We recorded the Lamplight-driven inhibitory response for a range of 405 nm intensities from 14.5–12.9 log photons/cm$^2$/s, 3.8−0.11 μW/mm$^2$ (Fig 2C), with change in RMP from $-42.1 \pm 0.5$ mV in dark to $-51.6 \pm 0.5$ mV after 5 s 405 nm stimulus (mean ± SEM). We were unable to detect changes in RMP at 12.6 log photons/cm$^2$/s, suggesting that 12.9 log photons/cm$^2$/s is close to threshold for opsin-driven activation. This is ~100-fold more sensitive to 1 s flashes than microbial opsins engineered for improved light sensitivity, such as gtACR2 (Govorunova et al, 2015).

Lamplight-induced silencing had two interesting properties. First, we found hyperpolarisation was stable once the 405 nm light was switched off (Fig 2D). Lamplight's sensitivity becomes even more marked when considering sustained inhibition, which requires repeated stimulation of many tools, including gtACR2. Even available step function tools (Cosentino et al, 2015; Alberio et al, 2018) are ~10-fold less sensitive than Lamplight. Second, in the majority of fluorescent cells inhibited by 405 nm light, we found hyperpolarisation could be reversed by subsequent presentation of 525 nm light (Fig 2D), to RMP of $-44.1 \pm 0.5$ mV (mean ± SEM, although some cells from injected SCN slices did not respond to 525 nm, Fig EV1C). For cells where inhibition could be reversed with 525 nm light, we observed this switching behaviour reliably across repeated stimulus presentations (Fig 2E). All SCN patching was conducted without addition of 9-cis retinal, indicating that Lamplight functions, and can achieve high sensitivity, without exogenous chromophore.

We next focused on the kinetics of responses to 405 nm and subsequent 525 nm light, which were fit with exponential curves to estimate response latency and velocity (Fig 2F). Onset of hyperpolarisation occurred rapidly to 405 nm with latency of $353.3 \pm 92.6$ ms and half-life of $273.9 \pm 54.2$ ms (Fig 2G and H, mean ± SEM). In comparison, it took several seconds for depolarisation by 525 nm to occur, with latency of $3.5 \pm 0.5$ s and half-life of $1.6 \pm 0.3$ s (mean ± SEM).

**Figure 2. Lamplight causes sustained reversible hyperpolarisation in SCN neurons.**

A Schematic of Lamplight AAV delivery to the suprachiasmatic nucleus (SCN) of $VIP^{Cre/+}$ mice. Whole-cell patching of SCN slices (bottom left) made by targeting mCherry fluorescent neurons (arrows, bottom right).

B Representative trace of hyperpolarisation to 405 nm light in a Lamplight-expressing SCN neuron.

C Hyperpolarisation to 405 nm was observed over a range of intensities. Recordings from 3 different cells across 2 SCN slices.

D Representative trace showing hyperpolarisation was sustained during dark after 405 nm light (5 s) and reversed by 525 nm light (15 s).

E Hyperpolarisation and depolarisation of a Lamplight-expressing neuron by 405 and 525 nm light, respectively, across multiple stimulus presentations.

F Three example traces from a single cell exposed to 405 nm followed by 525 nm light. Individual traces were fit with exponential decay (405 nm, blue) or association curves (525 nm, green) for 6 trials across 2 cells.

G, H Parameters from best-fit curves used to calculate (G) latency (ms) and (H) half-life (ms).

I Simultaneous 525 and 405 nm light prevents spike silencing.

J Constant 405 nm light (12.9 log photons) presented with 525 nm light of decreasing intensity from 14.9 to 12.9 log photons in 6 steps of 10 s (left) or 20 s (right). 525 nm light antagonised the 405 nm-driven hyperpolarisation for all intensities except 12.9 log photons.

Data information: For (C, G and H) data from different cells is indicated by different symbols. For (B, E and F) action potentials are truncated for clarity. Timing of 405 and 525 nm light indicated by blue and green horizontal bar. Intensities are log photons/cm$^2$/s.

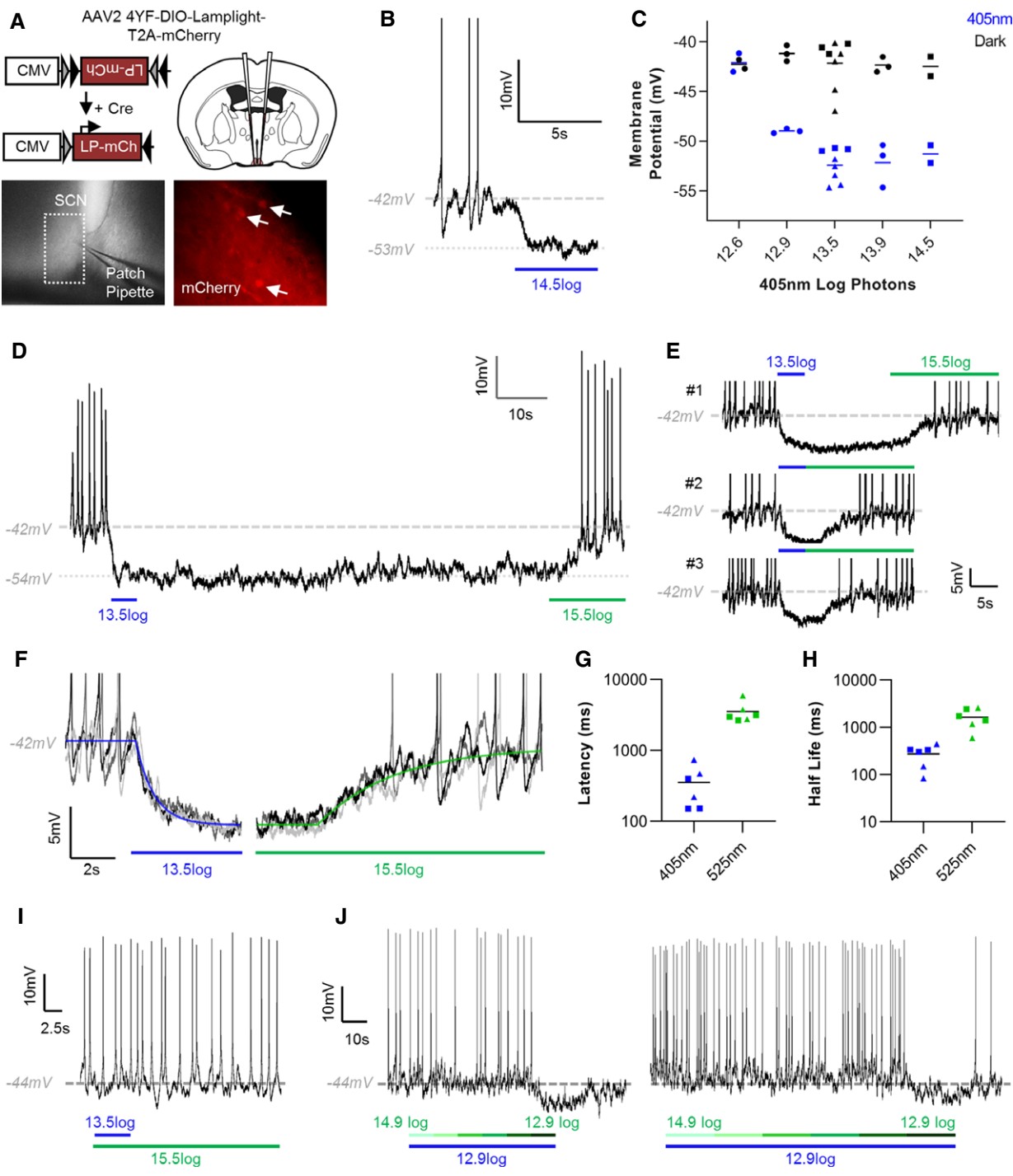

**Figure 2.**

Finally, we examined how neurons responded to simultaneous presentation of two wavelengths. 525 nm light antagonised the hyperpolarisation caused by 405 nm light, with no change in RMP observed when the two wavelengths were used simultaneously (Fig 2I), $-43.5 \pm 0.6$ mV, compared to preceding dark baseline, $-43.2 \pm 0.9$ mV (mean $\pm$ SEM). We found 15.5 log photons/cm$^2$/s or 22 μW/mm$^2$, of 525 nm light was sufficient to inhibit responses to both 13.5 and 14.5 log photons/cm$^2$/s 405 nm light, suggesting

neurons can also be controlled using Lamplight by modulating the spectral composition of light. To explore this further, we presented dim 405 nm light (12.9 log photons/cm$^2$/s) and superimposed 525 nm light of decreasing intensity across 6 steps (14.9–12.9 log photons/cm$^2$/s). Hyperpolarisation could only be observed for dimmest 525 nm light—suggesting inhibition is dependent on the ratio of 405 and 525 nm light (Fig 2J) and that Lamplight can be used as a tuneable inhibitory optogenetic tool to control neuronal firing.

## Using Lamplight to restore visual responses to the degenerate retina

The SCN data confirm Lamplight allows switchable neuronal inhibition. Theoretically, as light-induced changes in opsin state will be rapid, transitions could have high temporal resolution. However, the actual kinetics will be influenced by the downstream elements of the Gi/o cascade. Indeed, the response to 525 nm occurred over several seconds in SCN neurons. Finally, we applied Lamplight to a system in which temporal fidelity is a more critical feature of control to gain a better appreciation of its potential performance. In the intact vertebrate retina, visual signals are conveyed by G protein signalling with high temporal resolution at two stages—within the photoreceptor itself, and at the synapse between photoreceptors and ON bipolar cells. The ON bipolar cell response relies upon a Go cascade (Dhingra et al, 2002) and so represents an attractive target for Lamplight control.

We used intravitreal injection to introduce the Lamplight-T2A-mCherry AAV to L7$^{Cre/+}$ rd1 mice. These mice have rapid and extensive degeneration of rods and cones (Pittler & Baehr, 1991; Chang et al, 2002), and Cre recombinase expressed in Rod (ON) bipolar cells and a small subset of retinal ganglion cells (RGCs, Ivanova et al, 2010). As expected, we found transgene expression in the inner nuclear layer (location of ON bipolar cells) and a few RGCs (Fig 3A).

To assess the light response of Lamplight-expressing retinas, we used multi-electrode arrays to record RGC spiking in response to light impacting Lamplight expressed in the inner retina. We used 2 s of 405 nm (16.1 log photons/cm$^2$/s, 106 uW/mm$^2$), 20 s dark, then 2 s 525 nm (16.1 log photons/cm$^2$/s, 50 uW/mm$^2$). We observed three qualitatively distinct responses, not found in uninjected rd1 retinas (Fig EV2), leaving aside persistent increases in firing (Fig 3B), which could be produced by melanopsin (Tu et al, 2005; Weng et al, 2013; Eleftheriou et al, 2020). The most abundant was transient excitation to 405 nm (Fig 3C, top). A subset of RGCs with that response also increased firing to 525 nm. The other two classes had inhibitory responses to 405 nm but differed in temporal properties. In one (Fig 3C, middle), 405 nm induced a transient reduction in firing, with no clear response to 525 nm. In the other (Fig 3C, bottom), 405 nm induced sustained inhibition, lasting many seconds after the stimulus had ended and reproducibly reversed by

exposure to 525 nm. Such switching behaviour could be repeated for at least 20 stimulus repeats (Fig 3D).

A more detailed examination of the 3 response types (Fig 3E) revealed transient excitations to 405 nm on average built up over several seconds and had a mean response latency of 1.22 s (range 0.1–3.7 s). By contrast, inhibitory responses were more immediate with mean latency of 0.47 s (range 0.3–0.9 s) and 0.47 s (range 0.3–0.8 s) for transient and sustained responses, respectively. Excitatory responses to 525 nm were also relatively rapid, with mean response latency of 0.50 s (range 0.3–0.9 s).

We finally asked whether the amplitude of Lamplight responses in the retina could be controlled using the approaches employed in SCN neurons and HEK293 cells. To this end, we compared responses to 405 nm alone with those elicited by a 50:50 mix of 405 and 525 nm. We found that, indeed the mixed light produced a smaller response in all response classes. Thus, the degree of reduction in firing was smaller in cells with transient or sustained inhibition to 405 nm (mean ± SEM difference = 0.54 ± 0.22 spikes/s; one-sample t-test compared to 0, $t(13) = 2.47$, $P = 0.028$, Fig 3F), as was the increase in firing in cells excited by 405 nm (mean ± SEM difference = 1.2 ± 0.19 spike/s; one-sample t-test compared to 0, $t(33) = 6.47$, $P < 0.001$, Fig 3G).

Lamplight activation in the inner retina thus restores photosensitivity in degenerate retinas, producing a diversity of RGC response types consistent with the ability of retinal circuitry to invert polarity and apply temporal frequency filters during signal transfer and with other studies restoring photosensitivity to degenerate retinas (Cehajic-Kapetanovic et al, 2015; Gaub et al, 2015; Wyk et al, 2015; Berry et al, 2019). The behaviour of some RGCs confirm that 405 and 525 nm light support antagonistic changes in RGC firing within 500 ms. These characteristics could help recreate the high fidelity, analogue input required by the retina to support high spatiotemporal resolution vision—making Lamplight a potential candidate for vision restoration therapy.

## Conclusions

Here, we explored the potential of Lamplight, a bistable light-sensitive GPCR, as an inhibitory optogenetic tool. We found it can produce reversible hyperpolarisation in the brain and

---

**Figure 3. Lamplight expression in degenerate retinas can restore diverse and switchable responses to light.**

A  L7Cre rd1 retina expressing Lamplight-T2A-mCherry AAV. Anti-mCherry staining (white) was widespread (Left, scale bar = 500 μm). Fluorescent cell bodies found in different focal planes consistent with expression in ganglion (Top Right, scale bar = 50 μm) and bipolar cells (Bottom Right, scale bar = 50 μm).
B  Representative single unit showing characteristic ipRGC response.
C  Three Lamplight response categories—transient excitation (N = 35 biological replicates), transient inhibition (N = 6 biological replicates), and sustained inhibition (N = 8 biological replicates). Left panel shows representative single unit response to 2 s 405 or 525 nm light. Mean baseline normalised firing (grey shading shows SEM) is shown in centre left (for entire recording, 500 ms bins), centre right and far right panels (during 405 and 525 nm stimuli, respectively, 100 ms bins, N as above, except N = 5 biological replicates for sustained response to 525 nm).
D  Representative single unit showing Lamplight-driven light responses over 20 stimulus repeats.
E  Distribution of response latencies for the three different types of Lamplight-driven responses to 405 nm, as well as reversal caused by 525 nm in sustained inhibition units. Two outliers with slow latency (> 2.5 s) in transient excitation group are likely responding to stimulus offset (N = 35 for Transient Excitation, N = 6 for Transient Inhibition, N = 7 for Sustained Inhibition and N = 5 for Sustained Excitation, all biological replicates).
F  Inhibition responses were smaller and less sustained for mixed stimuli (50% 405 nm, 50% 525 nm) compared to 405 nm only (N = 14 biological replicates).
G  Excitation responses were attenuated for mixed stimuli compared to 405 nm only (N = 35 biological replicates).

Data information: For (F–G) Left panel shows representative single unit. Right panel shows distribution of difference in firing rate for 405 nm versus mixed stimulus. All representative single units show perievent rasters (first trial at top) and firing rate histograms (Bin size = 500 ms). Timing of light stimuli shown by shaded vertical bars. All light stimuli are 16 log total Lamplight effective photons/cm$^2$/s. Error bars show SEM. *$P < 0.05$, ***$P < 0.001$ (One-sample t-test compared to zero).

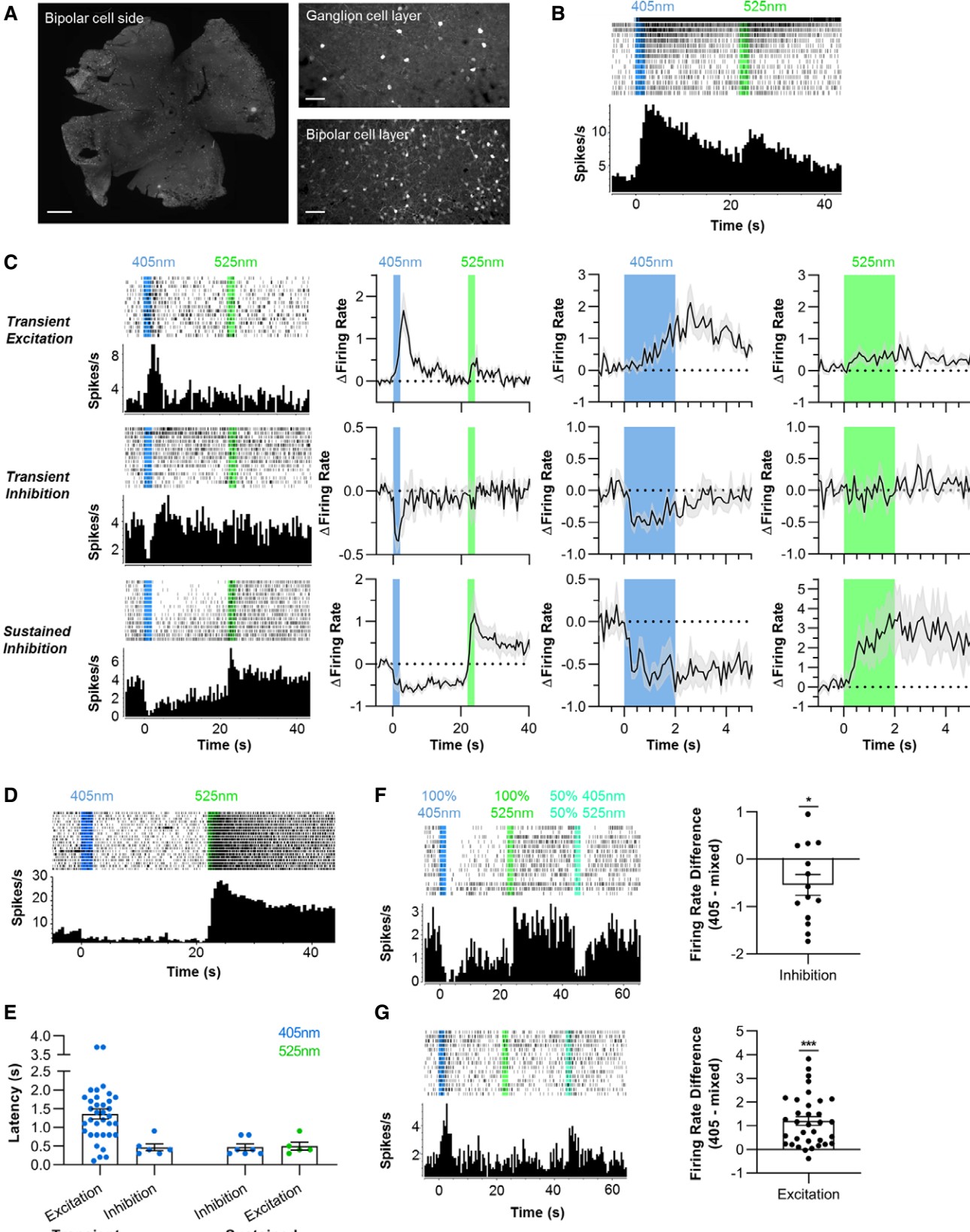

**Figure 3.**

photo-switchable changes in the retina. In both *in vivo* systems, single light pulses could produce sustained inhibition. The bistable nature of Lamplight allows its degree of activity to be controlled by modulating either the intensity or spectral composition of the light stimulus. Our data establish bistable animal opsins as promising high sensitivity optogenetic tools for switchable neuronal inhibition.

Lamplight compares favourably with available inhibitory optogenetic tools. It is sensitive to low intensity light, reducing the requirement for potentially damaging levels of light and heat, and addressing the challenge of controlling widely dispersed neuron populations. Lamplight-driven effects on activity could be measured for tens of seconds after a 405 nm pulse, before being switched off by 525 nm. Such prolonged inhibition would require repeated illumination with high intensity light for most published inhibitory tools, although some, such as SwiChR (Berndt *et al*, 2014), can achieve similar long lasting activity, albeit with reduced light sensitivity. Light-induced transitions between active and inactive states for Lamplight (as all opsins) should be fast, but hysteresis in the G protein cascade induces delays in the cellular response. Such delays will invariably be longer than for microbial opsins that more directly control ionic conductance. While Lamplight-driven responses to 405 nm occurred within a few hundred milliseconds in both SCN and retinal neurons, deactivation following 525 nm was more sluggish in the SCN, compared to the retina where RGC firing changed within 500 ms. This difference plausibly reflects intrinsic properties of the Gi/o signalling cascades of the different cell types.

Lamplight could potentially be widely used. Gi/o signalling is a ubiquitous cellular mechanism (Syrovatkina *et al*, 2016), employed by numerous native GPCRs for neuronal inhibition, including opioid and dopamine receptors (Lüscher & Slesinger, 2010). In particular, Go is widely expressed in the central nervous system (CNS, Jiang & Bajpayee, 2009). CNS disorders are amongst the most abundant targets for GPCR drug discovery (Hauser *et al*, 2017), with Gi/o-coupled receptors involved in schizophrenia, Parkinson's disease, and Alzheimer's disease. While Lamplight may conceivably also require GIRK expression for inhibition, experience with Gi/o chemogenetic tools suggests that this is not a concern (Roth, 2016). In future, a chimera between Lamplight and non-photosensitive GPCRs could tune G protein specificity and further engage native receptor signalling (Tichy *et al*, 2019).

In addition to being Gi/o coupled, we chose Lamplight over other bistable animal opsins because of its spectral sensitivity. Lamplight has clear separation between the wavelength of peak sensitivity ($\lambda_{max}$) of its inactive and active states. This is critical in allowing effective activation and deactivation. Moreover, the long-wavelength shift of Lamplight's active state is an important characteristic. All opsin photopigments (irrespective of $\lambda_{max}$) retain significant sensitivity to short wavelengths (Partridge & De Grip, 1991). For this reason, long wavelengths can theoretically drive the vast majority of Lamplight to an inactive state, whereas shorter wavelengths drive a more balanced equilibrium of active and inactive opsin. Conversely, in other opsins with a short wavelength-shifted active state, long wavelengths may very effectively activate, but short wavelengths would not fully inactivate. The Lamplight scenario is preferential both because effective silencing is important in its own right and because this characteristic allows greater signal:noise. Perhaps the biggest disadvantage with Lamplight is that it is most effectively activated by UV light ($\lambda_{max}$ = 370 nm). We show here that this does not preclude activation by > 400 nm light at intensities at least 10-fold lower than other inhibitory tools, but a pigment shifted to longer wavelengths could enhance sensitivity across a greater range of visible wavelengths.

In conclusion, Lamplight demonstrates that bistable animal opsins can provide efficient inhibitory optogenetic tools that can be used for sustained and reversible neuronal silencing. Lamplight allows optogenetic control that is physiological and can be tuned by altering intensity and spectral composition of light stimulus. The unique properties of Lamplight make it well suited to basic research, as well as therapeutic applications.

# Materials and Methods

### Cell culture

Hek293T cells (ATCC) were incubated at 37°C (5% $CO_2$) in culture media (Dulbecco's modified Eagle's medium with 4,500 mg/l glucose, L-glutamine, sodium pyruvate and sodium bicarbonate from Sigma) with penicillin (100 U/ml), streptomycin (100 μg/ml) and 10% foetal bovine serum (FBS). Cell were seeded into 12-well plates at a density of 250,000 cells/well in antibiotic-free culture medium. After 48 h, cells were transiently transfected using Lipofectamine 2000 (Thermo Fisher) according to manufacturer's instructions.

For BRET G protein activation assay, each well of 12-well plate was transfected with following: 100 ng sVβ1, 100 ng sVγ2, 25 ng mGRK3-nLuc, 50 ng Gαo and 500 ng opsin (where applicable). BRET assay plasmids were obtained from Kiril Martemyanov & Ikuo Masuho (Scripps Research Institute). Gαo was used as it is the most abundantly expressed member of the Gi/o/t family (Jiang & Bajpayee, 2009). All subsequent steps were conducted under dim red light. After addition of transfection reagent and DNA, cells were incubated for 4–6 h at 37°C and then resuspended in 1ml of culture media containing 10 μM 9-*cis* retinal. 100 μl of cell suspension was added to each well of a white-walled clear-bottomed 96-well plate (Greiner Bio-One) and left overnight before performing BRET G protein activation assay.

For immunocytochemistry, cells were plated onto coverslips and transfected with plasmid as described above. After 24 h, cells were fixed using 4% paraformaldehyde (PFA) in PBS and washed with PBS. All dilutions were carried out in PBS with 0.05% Tween-20. Cells were first blocked in 5% normal donkey serum and then incubated in 1:500 dilution of primary antibody (mouse anti-1D4, Abcam, catalogue number ab5417) with 1% donkey serum for 1 h at room temperature. Cells were washed three times and then incubated in 1:500 dilution of secondary antibody (Donkey anti-mouse 488, Abcam, catalogue number A-210202) for 30 min at room temperature in the dark. Cells were washed four times, then mounted onto slides using Prolong Anti-fade Gold media with DAPI and allowed to dry overnight at room temperature in dark.

### BRET G protein activation assay

All following steps were carried out under dim red light. Before beginning the BRET G protein activation assay, culture media was removed from cells and replaced with 50 μl imaging media (L-15 media without Phenol Red containing L-glutamine (Gibco), 1% FBS

and 10 μM 9-cis retinal). Cells were then left to incubate at room temperature in dark for up to 2 h. Under dim red light, NanoGlo Live Cell substrate (Furimazine derivative, Promega) was diluted 1:40 in PBS. 12.5 μl of dilute NanoGlo substrate solution was added to each well of 96-well plate (final dilution of 1:200), for up 6 wells at a time, and incubated for 5 min to allow luminescence to equilibrate. BRET measurements were conducted using a FluoStar Optima microplate reader (BMG Labtech). As this plate reader has a single photomultiplier tube (PMT), light emitted by fluorescent Venus and bioluminescent Nanoluc were measured sequentially using 535 nm (30 nm FWHM with gain set to 4095) and 470 nm (30 nm FWHM with gain set to 3,600) emission filters. A 0.68 s recording interval was used for each filter, with a total cycle time of 2 s. We coupled the microplate reader bottom optic to the liquid light guide of a pE-4000 light source (CoolLED) to allow us to flash cells inside the microplate reader. To avoid bleaching the PMT during light stimulus, a motorised shutter was built to protect the PMT while light was on. The activity of this shutter was synced to the light source using an Arduino microcontroller. To avoid neighbouring wells being exposed to light, each recorded well was surrounded by empty wells and the location of wells measured was counterbalanced. For every recording, we measured five cycles of baseline measurement (10 s total) and then flashed cells with 1 s of light before resuming recording for up to 45 cycles (90 s total). For repeated light exposure, sequential recordings of the same well were performed. Timing of light stimuli and recording were controlled using plate reader software (Optima in script mode). A range of different light stimuli was used (all 1 s in duration), described in more detail in Table EV1. The spectral power distribution of the stimuli was measured using a SpectroCAL MKII spectroradiometer (Cambridge Research Systems Ltd).

## Animals

All experiments were conducted in accordance with the UK Animals (Scientific Procedures) Act (1986). $L7^{Cre/+}$ (Marino et al, 2002) rd1 transgenic mice on a mixed C3H × C57Bl/6 background were used for retinal MEA experiments. L7Cre mice express Cre recombinase in rod bipolar cells and a subset of retinal ganglion cells (Ivanova et al, 2010). They also possess the $Pde6b^{rd1}$ mutation (Pittler & Baehr, 1991; Chang et al, 2002), which causes progressive retinal degeneration, with vision loss complete once animals are over 80 days old. L7Cre rd1 mice were genotyped to confirm they do not possess the GRP179 point mutation (Peachey et al, 2012) which affects bipolar cell function. $VIP^{Cre/+}$ mice (Taniguchi et al, 2011) on a mixed C57BL/6 × 129S4Sv/Jae background were used for SCN patch electrophysiology experiments. These mice possess Cre recombinase in vasoactive intestinal polypeptide (VIP)-expressing cells. All mice were kept under a 12:12 light–dark cycle with food and water provided ad libitum.

## AAV virus

$L7^{Cre/+}$ rd1 mice received bilateral intravitreal injections of lamprey parapinopsin, termed Lamplight, virus (AAV2 4YF– ITR - DIO-CMV-LPPN-1D4-T2A-mCherry – WPRE- SV40 late polyA - ITR). The Lamplight virus was packaged in an AAV2/2 capsid with four tyrosine to phenylalanine mutations (Petrs-Silva et al, 2011) to achieve

efficient viral transduction of retinal cells, in particular bipolar cells. An 8 amino acid C-1D4 tag (ETSQVAPA) was added to the C-terminus of the lamprey parapinopsin transgene (AB116380.1). The Lamprey parapinopsin-1D4 transgene (LPPN-1D4) was linked to a mCherry fluorescent reporter using a T2A sequence to ensure 1:1 co-expression of the two proteins. The inverted LPPN-1D4-T2A-mCherry open reading frame was flanked by two pairs of Lox sites (LoxP and Lox2272), so that in the presence of Cre recombinase, the transgene is inverted into the sense orientation and expression is driven by the constitutive CMV (cytomegalovirus) promotor. A woodchuck hepatitis virus post-transcriptional regulatory element (WPRE) and SV40 late polyA sequence were also included between ITRs to improve transgene expression. Virus was obtained from VectorBuilder.

## Brain injections

Heterozygous $VIP^{Cre/+}$ mice were injected aged 10–11 weeks old using previously established protocol (Paul et al, 2020). Briefly, mice were anaesthetised using 1% isoflurane and positioned in a stereotaxic frame. Craniotomy and bilateral AAV injection was carried out using a computer-controlled motorised injector (Drill and 473 Microinjection Robot, Neurostar), and 69 nl of Lamplight-T2A-mCherry virus was injected into SCN at 25 nl/s. Micropipettes were left in place for 5 min after injection and then removed in 100 μm/5 s increments until outside SCN.

Mice were culled by cervical dislocation during the light phase (ZT4-8) 6 weeks after SCN injection. Brain slices were prepared as described previously (Belle & Piggins, 2017; Hanna et al, 2017). Briefly, brains were removed and mounted on metal stage. Coronal slices containing mid-SCN levels (250 μM) across the rostro-caudal axis were cut using a Campden 7000smz-2 vibrating microtome (Camden Instruments). Slicing was in ice-cold (4°C) sucrose-based incubation solution containing: 3 mM KCl, 1.25 mM $NaH_2PO_4$, 0.1 mM $CaCl_2$, 5 mM $MgSO_4$, 26 mM $NaHCO_3$, 10 mM D-glucose, 189 mM sucrose, oxygenated with 95% $O_2$, 5% $CO_2$. After cutting, slices were left to recover at room temperature in a holding chamber with continuously gassed incubation solution for at least 20 min before transferring into artificial cerebrospinal fluid (aCSF). aCSF composition was the following: 124 mM NaCl, 3 mM KCl, 24 mM $NaHCO_3$, 1.25 mM $NaH_2PO_4$, 1 mM $MgSO_4$, 10 mM D-Glucose, 2 mM $CaCl_2$ and 0 mM sucrose.

## Whole-cell current-clamp recordings

Coronal brain slices containing mid-level of the SCN were placed in the bath chamber of an upright Leica epi-fluorescence microscope (DMLFS; Leica Microsystems Ltd) equipped with infrared video-enhanced differential interference contrast (IR/DIC) optics. Slices were held in place using a tissue anchor grid and continuously perfused with aCSF by gravity (~2.5 ml/min) without addition of exogenous 9-cis retinal. Brightfield photographs of the patch pipette sealed to these neurons were taken at the end of each recording for accurate confirmation of the anatomical location of the recorded cell within the SCN.

Patch pipettes (resistance 7–10 MΩ) were fashioned from thick-walled borosilicate glass capillaries (Harvard Apparatus) pulled using a two-stage micropipette puller (PB-10). Recording pipettes

were filled with an intracellular solution containing the following: 130 mM K-gluconate, 10 mM KCl, 2 mM MgCl$_2$, 2 mM K2-ATP, 0.5 mM Na-GTP, 10 mM HEPES and 0.5 mM EGTA, pH adjusted to 7.3 with KOH, measured osmolarity 295–300 mOsmol/kg). Cell membrane was ruptured under minimal holding negative pressure.

An Axopatch Multiclamp 700A amplifier (Molecular Devices) was used in a current-clamp mode with no holding current (I = 0), after establishing whole-cell configuration at −70 mV. Signals were sampled at 25 kHz and acquired in gap-free mode using pClamp 10.7 (Molecular Devices). Access resistance for the cells used for analysis was < 30 MΩ. All data acquisition and protocols were generated through a Digidata 1322A interface (Molecular Devices). All recordings were performed at room temperature (~23°C).

mCherry fluorescent SCN neurons were identified with a 40× water immersion UV objective (HCX APO; Leica) using a cooled Teledyne Photometrics camera (Retiga Electro). Red fluorescence was detected using 550 nm excitation light from pE-4000 light source (CoolLED) and a Leica N2.1 filter cube (Dichroic = 580 nm, Emission filter = LP 590 nm). The same light source was used to deliver 405 and 525 nm light stimuli via the 40× UV objective. Patching data were analysed using Clamplit 10.7, Prism and Excel. Resting membrane potential was measured manually in pClamp for 10 s before stimulus and at end of 405 nm stimulus. For kinetics analysis, data were first downsampled (extract every 100$^{th}$ data point, new sample frequency = 4 ms). For responses to 405 nm, data from first 3 s after stimulus onset were fit with a plateau followed by either an exponential decay or exponential association curve. For responses to 525 nm, we used data from first 15 s after stimulus onset. We excluded any data > 1 standard deviation from mean (to remove spikes that might distort fit) and then fit with plateau followed by exponential association. Non-linear curve fitting was performed using least-squares minimisation in Excel. Data from 2 replicates were excluded due to poor curve fits ($R^2$ < 0.3).

## Intravitreal injections

Heterozygous $L7^{Cre/+}$ rd1 mice were injected aged 9–10 weeks. Mice were anaesthetised by intraperitoneal injection ketamine (75 mg/kg body weight) and medetomidine (1 mg/kg body weight). Once anaesthetised, mice were positioned on a heat mat to prevent cooling. Pupils were dilated with 1% tropicamide eye drops (Bausch & Lomb) and a 13 mm coverslip was positioned on gel lubricant (Lubrithal) applied to the cornea. Approximately 2.2 μl of virus (1.12 × 10$^{13}$ genomic counts per ml) was injected into the vitreous of each eye using a Nanofil 10-μl syringe (World Precision Instruments) using 35-gauge bevelled needle using a surgical microscope (M620 F20, Leica). All mice received bilateral injections. Anaesthesia was reversed by intraperitoneal injection of atipamezole (3 mg/kg body weight). During recovery, 0.5% bupivacaine hydrochloride and 0.5% chloramphenicol were applied topically to the injected eyes. Mice also received 0.25 ml of warm saline by subcutaneous injection to aid recovery.

## Multi-electrode array recordings

Multi-electrode array recordings were conducted between 13 to 15 weeks after bilateral intraocular injection. Mice were dark-adapted overnight. All following steps were performed under diffuse dim red light. Dark-adapted mice were culled by cervical dislocation (approved Schedule 1 method). Enucleated eyes were placed in petri dish filled with carboxygenated (95% O$_2$/ 5% CO$_2$) Ames' media (supplemented with 1.9 g/l sodium bicarbonate, pH 7.4, Sigma-Aldrich) and retinas dissected, with care taken to remove vitreous from inner retinal surface. Retinal wholemounts were then placed on glass coated metal harps (ALA Scientific Instruments) and positioned ganglion-cell side down on coated 256-channel multi-electrode arrays (MEA, Multi Channel Systems).

Multi-electrode arrays were first incubated in foetal bovine serum overnight at 4°C and then coated with 0.1% polyethyleneimine (PEI) in borate buffer (pH8.4) for 1 h at room temperature. PEI coating was then removed, and MEA washed 4–6 times with ddH$_2$0. PEI-coated MEAs were then air-dried and coated with 20 μg/ml laminin in fresh Ames' medium for 30–45 min at room temperature (Lelong *et al*, 1992; Egert & Meyer, 2005). Laminin solution was removed before retina was positioned on the MEA. Once in place on the MEA, the retina was continuously perfused with carboxygenated Ames' media with 10 μM 9-*cis* retinal at 2–3 ml/min using a peristaltic pump (PPS2, Multi Channel Systems) and maintained at 34°C using a water bath heater (36°C), in-line perfusion heater (35°C) and base plate heater (34°C). Once positioned, retinas were perfused in dark for at least 45 min before first light stimuli were applied.

Data were sampled at 25 kHz using MC Rack software (Multi Channel Systems). A Butterworth 200 Hz high pass filter was applied to raw electrode data to remove low frequency noise. Amplitude threshold for spike detection was 4–4.5 standard deviations from baseline. Light stimuli were presented using a customised light engine (Thorlab LEDs). An Arduino Due microcontroller controlled by programmes written in LabVIEW (National Instruments) to control stimulus duration and intensity by altering LED output. Table EV1 shows the intensity of light stimuli used for testing Lamplight activity.

Multi-unit data were spike sorted into single units using Plexon Offline Sorter (using Template Sorting Method and Principal Component Analysis, followed by manual verification) and then analysed using Neuroexplorer, Matlab 2012 and Prism. Single units with light responses were identified using following approach. A perievent stimulus time histogram (PSTH) was generated for first 15 trials with 500 ms bin size. Lamplight-driven light responses were identified as significant change in spike firing rate within 5 s of 405 nm stimulus onset—defined as 2 standard deviations from mean baseline (5 s before 405 nm stimulus onset) for at least 2 bins (1 s). Units were excluded if they had insufficient spiking to provide meaningful baseline spiking rate (defined as spiking in < 50% baseline bins), had square wave shape caused by removal of noisy spikes during spike sorting, or responses that were not reproducible across trials (15 trials were split into groups of 3, then PSTH compared to confirm reproducibility of responses). The remaining 52 units were then divided into categories based on their response profile. For latency, we used data in 100 ms bins and identified first bin after light stimulus where data were more than 2 standard deviation from baseline (5 s before stimulus onset). For comparing responses from mixed and 405 nm only stimuli, we used data in 500 ms bins. For transient excitation responses, we compared difference in average spike firing rate for first 5 s after stimulus onset. For inhibition responses, we compared difference in average spike firing rate across interstimulus interval (20 s total). A one-sample *t*-

test was used to compare difference values relative to zero (no change). Data were normally distributed according to D'Agostino–Pearson normality test—for Inhibition, K2 = 0.99, $P$ = 0.61, and for Excitation, K2 = 3.53, $P$ = 0.17.

### Immunohistochemistry

Injected L7Cre mice were culled by cervical dislocation (approved schedule 1 method). Eyes were removed and placed in 4% paraformaldehyde in phosphate-buffered saline for 24 h at 4°C. Retinas were then dissected and permeabilised in 1% Triton X in PBS for 3 × 10 min at room temperature. Retinas were then blocked using 1% Triton X in PBS with 10% normal goat serum for 2–3 h while shaking gently. Retinas were incubated in primary antibody (1:500 dilution of rabbit anti-mCherry antibody, Kerafast, catalogue no. EMU106, in 1% Triton X in PBS with 2.5% goat serum) overnight at room temperature. Retinas were washed in PBS with 0.2% Triton X for 4 × 30 min shaking gently. Retinas were incubated in secondary antibody (Goat anti-rabbit Alexa fluorophore 555, Abcam, catalogue no. ab150078, in 1% Triton X in PBS with 2.5% goat serum) for 3–4 h at room temperature in the dark. Retinas were washed a further four times in PBS with 0.2% Triton X for 30 min each and then washed in ddH$_2$0 for 10 min. Retina was then mounted on a microscope slide using Prolong Gold antifade mountant and allowed to dry overnight at room temperature.

### Fluorescence microscopy

Images of retinal wholemount were acquired using a Leica DM2500 microscope with DFC365 FX camera (Leica) and a CoolLED-pE300-white light source. Imaging software was Leica Application Suite Advanced Fluorescence6000. Images were acquired using Chroma ET Y3 (excitation = 545 nm, emission = 610 nm), A4 (excitation = 360 nm, emission = 470 nm) and L5 (excitation = 480 nm, emission = 527 nm) filter sets. Images of different retinal focal planes were acquired on an Andor Dragonfly200 spinning disk upright confocal microscope (with a Leica DM6 FS microscope frame). Samples where excited using 561 nm diode laser via Leica RFP filter (excitation = 546 nm, emission = 605 nm). Images were collected using a Zyla 4.2 Plus sCMOS. Global enhancements to image brightness and contrast were made using ImageJ software.

### Generating ratiometric bispectral light stimuli

Spectra of monochromatic light stimuli (405 and 525 nm) were measured using a SpectroCAL MKII spectroradiometer (Cambridge Research Systems, Ltd) from 380 to 750 nm. Spectra were converted from W/m$^2$ to Photons/cm$^2$/s. Log effective photons for inactive state and active state for each wavelength were calculated by multiplying light spectra by log relative sensitivity of opsin at each wavelength. Opsin nomograms for Lamplight inactive and active state were generated with $\lambda_{max}$ of 370 and 515 nm (Koyanagi et al, 2004), respectively, using A1-visual pigment template (Govardovskii et al, 2000). In addition to the effective photons for the inactive or active state, the total effective photons for Lamplight (i.e. light that either state could respond to) were calculated by summing effective photons for both states. Throughout text, unless specified otherwise, intensities provided are in total Lamplight effective photons.

To generate ratiometric stimuli, the total Lamplight effective photons for 405 nm only or 525 nm only stimuli were first matched. Take 15.5 log total Lamplight effective photons/cm$^2$/s. For the 405 nm only stimulus, this would result in 15.29 log inactive state effective photons and 15.09 log active state effective photons, 60% of total effective photons are for inactive state, while remaining 40% are for active state. For the 525 nm only stimulus, 15.5log total Lamplight effective photons would mean 12.18 log inactive effective photons/cm$^2$/s and 15.5 log active state effective photons/cm$^2$/s, 100% of total effective photons are for active state, with 0% for inactive state. These matched values are then used to define ratiometric bispectral stimuli on a scale of 1 to 0 relative amount of 405 nm light in the stimulus (where 1 = 405 nm light only or maximum possible R effective photons and 0 = 525 nm light only minimum possible inactive state effective photons). Continuing the example above, for 15.5 log total Lamplight effective photons/cm$^2$/s, a stimulus with 0.25 relative 405 nm light in stimulus, would require 14.68 log inactive state effective photons/cm$^2$/s and 15.43 log active state effective photons/cm$^2$/s, where 15% of total effective photons are for inactive state and 85% of total effective photons are for active state (0.15/0.6 = 0.25 of maximum inactive state effective photons).

## Data availability

Data have not been deposited in an external public repository but can be shared on reasonable request to the corresponding author.

**Expanded View** for this article is available online.

### Acknowledgements
This work was funded by grants from Human Frontier Science Program (RGP0034/2014) and Medical Research Council (MR/N012992/1 and MR/S026266/1) awarded to RJL. POF was funded by the Bekker Programme (PPN/BEK/2018/1/00192) implemented by the Polish National Agency for Academic Exchange. RS was funded by David Sainsbury Fellowship (NC/P001505/1) from National Centre for the Replacement Refinement & Reduction of Animals in Research. NM funded by Fight for Sight Fellowship (5047/5048). AEA funded by Dean's Prize Fellowship from The University of Manchester. MDCB funded by grant from Biotechnology and Biological Sciences Research Council (BB/S01764X/1).

### Author contributions
JR, RJL, NM, ERB and AA designed the experiment. JR, BB-O, MDCB, SP, RH, PW, RM, NM, POF, FM, JW, RS, AEA, and TB acquired and analysed the data. JR and RJL wrote the manuscript. All authors approved the submitted version.

### Conflict of interest
JR and RJL are named inventors on a patent application using Lamplight to control G protein signalling (PCT/GB2019/052685).

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
