## [Review Process File · EMBO Reports]

Using a bistable animal opsin for switchable and scalable optogenetic inhibition of neurons

Jessica Rodgers, Beatriz Bano-Otalora, Mino Belle, Sarika Paul, Rebecca Hughes, Phillip Wright, Richard McDowell, Nina Milosavljevic, Patrycja Orłowska-Feuer, Franck Martial, Jonathan Wynne, Edward Ballister, Riccardo Storchi, Annette Allen, Timothy Brown, and Robert Lucas
DOI: 10.15252/embr.202051866

Corresponding author(s): Jessica Rodgers (jessica.rodgers@manchester.ac.uk), Robert Lucas (robert.lucas@manchester.ac.uk)

Review Timeline:	Transfer from Review Commons:	9th Oct 20
	Editorial Decision:	24th Nov 20
	Revision Received:	29th Jan 21
	Accepted:	2nd Feb 21

Editor: Esther Schnapp

Transaction Report: This manuscript was transferred to EMBO reports following peer review at Review Commons

Review #1

1. How much time do you estimate the authors will need to complete the suggested revisions:

Estimated time to Complete Revisions (Required)

(Decision Recommendation)

Less than 1 month

2. Evidence, reproducibility and clarity:

Evidence, reproducibility and clarity (Required)

SUMMARY:

Rodgers and Lucas and team demonstrate a new and powerful method for the optical regulation of neuronal activity in optogenetic experiments. This method consists of the heterologous expression of the parapinopsin protein from Lamprey, which is a bistable and predominantly G α -i/o-coupled opsin. This opsin is functionally characterized in a signaling assay in HEK293 cells and then for control of action potentials in SCN central neurons and retinal bipolar cell neurons. Key strengths of this opsin over other opsins are bistability (the ability to persist in the active state upon withdrawal of light, and the ability to terminate this state using a second colour of light) and an exceptional light sensitivity, and these are convincingly demonstrated in neuronal applications here.

MINOR COMMENTS:

I only have minor comments on control experiments and manuscript clarity.

Controls:

1)The authors apply three separate assays to demonstrate parapinopsin function in each with very good outcomes. Control experiments in which the examined cell types are exposed to light in the absence of the opsin appear to be missing. It would be desirable to add this data, which the authors certainly have already collected, to the manuscript.

2)Is addition of the rhodopsin-derived 1D4 epitope, which may contain phosphorylation sites, expected to impact parapinopsin function or expression?

3)What is the maximal number of switching cycles ON/OFF that the authors have achieved using this opsin?

Clarity:

1)On page 5 the authors state: "suggesting varying ratio of 405nm and 525nm light will lead to different degrees of hyperpolarization (Figure 2J)" - I am not convinced that Figure 2J is the dataset that supports this specific point.

2)To assist with adaptation of this opsin in other laboratories, it could be important to clarify in which experiments exogenous 9 cis-retinal was added, and more generally in which types of experiments this will be recommended/required.

3) On page 2, 2nd paragraph, middle: It may be pointed out that deactivation of other opsins is not only outside of the experimenter's control but often also rapid (which is undesirable if single-shot inhibition is required).

4) On page 8, the authors state that this opsin combined with ratiometric stimulation may perform better than others when controlling a population of neurons with varied opsin expression; I am not sure if this is accurate: wouldn't ratiometric stimulation result in a greater or a smaller number of active opsins in cells that express higher or lower opsin levels, and thereby lead to different outcomes across the population?

5) It would be desirable to express light intensities for some of the most important experiments also in wattage per area, which appears to be a common representation in the field.

6) The authors cite the relevant literature in the field; perhaps one work of the Bruchas laboratory (Siuda Neuron 2015) and one of the Isacoff laboratory (Janojak Nature Neuroscience 2009) could be additionally included and discussed, because these are relevant references for either the opiod-opsin chimeras mentioned on page 2 (Siuda) or for a bistable light-sensitive channel for neuronal inhibition (Janoiak).

7) Minor improvements to the text could be considered, e.g. 10x => 10-fold (abstract), 405nm and 525nm => 405 and 525 nm, "may be used to switchable control" => "may be used to OBTAIN/ALLOW/ACHIEVE switchable control", "1s flashes of light" is repeated on page 7.

3. Significance:

Significance (Required)

This study will have an exceptional impact on the field of optogenetics and neuroscience. Tools for reliable action potential silencing, in particular at low light levels, are not generally available. The data presented by the authors show clearly that this is possible with the lamprey parapainopsin through a native-like G α -coupling mechanism. This study goes beyond this primary observation by demonstration of the additional benefit of bistability, and emerging from this, the ability to tune activity levels by light composition, making this opsin a potential future go-to method in the field.

Review #2

1. How much time do you estimate the authors will need to complete the suggested revisions:

Estimated time to Complete Revisions (Required)

(Decision Recommendation)

Between 1 and 3 months

2. Evidence, reproducibility and clarity:

Evidence, reproducibility and clarity (Required)

The authors demonstrate the usability of the Gi/o-coupled, bistable animal opsin Lamplight for neuronal inhibition. Lamplight can be switched between signaling active and inactive states by the application of near UV- and green light pulses respectively, whereby the signaling active state triggers neuronal hyperpolarization via activation of the G-protein coupled inwardly rectifying potassium channel (GIRK).

The authors show Lamplight-mediated, reversible suppression of spontaneous neuronal activity in the hypothalamic suprachiasmatic nucleus. Thereby Gi/o-mediated optogenetic silencing of neurons could be accomplished at comparatively low light intensities. Moreover, the authors show that Lamplight activation in mainly on-bipolar cells of rd1 mice, serving as a mouse model of retina degeneration, results in responses of opposite signs in retinal ganglion cells.

There are a few major concerns:

The authors seem to ignore relevant work of the Kleinlogel and Roska labs on optogenetic vision restoration. Moreover, statements on kinetics need to be balanced also in the abstract: "rapid" seems incompatible with sec time courses (see also below).

p.2, second paragraph

This paragraph compares bistable animal opsins with non-bistable animal opsins. The optogenetic inhibition by Lamplight is described as "rapidly reversible" in the abstract. This statement should be corroborated in this paragraph by addressing the following question: How fast is the deactivation of Lamplight, when the deactivation process is accelerated by green light illumination, in comparison to the solely light-off switch triggered deactivation of other animal opsins like e.g. MW opsin (Berry et al., 2019, Eickelbeck et al., 2020)? Alternatively, optogenetic silencing should be described as "reversible" instead of "rapidly reversible" in the abstract.

p. 3 lines 38-40

"... this time course likely reflects the time taken for decay of the GRK3:beta-gamma complex, as light-induced changes in opsin conformation are near instantaneous. "

The authors state that the time course of the BRET signal likely reflects the kinetics of the GRK3:beta-gamma complex formation/decay. Can the additional deactivation of highly light sensitive Lamplight molecules by BRET assay derived light sources, namely the light which is emitted by bioluminescent Nanoluc ($\lambda = 470$ nm) and fluorescent venus ($\lambda = 535$ nm), be ruled out? An additional deactivation due to the assay conditions would influence the time course of the BRET signal. Is the kinetics of the BRET signal independent of the recording times/intervals? If not, the potential influence of the assay conditions on the measured kinetics should be discussed.

p. 5 lines 20-24

"Onset of hyperpolarization occurred rapidly to 405 nm with mean latency of 353.3 ± 92.6 ms and half-life of 273.9 ± 54.2 ms (Figure 2G-H, mean \pm SEM). In comparison, it took several seconds for depolarisation by 525nm to begin, with mean latency of 3.5 ± 0.5 s, and responses were much slower, with average half-life of 1.6 ± 0.3 s (mean \pm SEM)."

Please compare the kinetics of neuronal inhibition and disinhibition to the BRET signal derived kinetics which likely reflect the formation/decay of the GRK3:beta-gamma complex (Fig.1). If known, please indicate which steps in the Gi/o pathway downstream of Lamplight activation and upstream of GIRK activation are rate limiting?

p.6 lines 37-40

"The diversity of RGC responses to light in the Lamplight retina is consistent with the established ability of the retinal circuitry to invert polarity and apply different temporal frequency filters during signal transfer, and with the outcome of other studies restoring photosensitivity to ON bipolar cells. "

The demonstration of the usability of Lamplight for neuronal inhibition in the retina is impeded by the complexity of the system, in particular because the electrophysiological measurements were solely performed on RGCs. The impact of the paper would be substantially improved, if additional electrophysiological measurements were performed on Lamplight expressing bipolar cells while different light stimulation protocols are applied.

p.7 line 46 - p.8 lines 1-2

"Such 'ratiometric' control could be a useful complement to the standard method of controlling opsin activation by altering light intensity, as it has the potential to provide a more uniform level of activity across a population of neurons differing in the amount of opsin expressed, and lying at different distances from the light source."

Only UV-light activated Lamplight can be deactivated by additional green light illumination. The number of activated Lamplight molecules still depends on opsin expression and the distance from the light source. Why is the level of activity then considered being more uniform?

p. 8 lines 36-38

"Opsin photopigments have a stereotyped spectral sensitivity profile that is characterized by a much more complete loss of sensitivity at wavelengths longer vs shorter than the peak."
This is true for groundstate activation. Please provide references, which show that this is also true for light activated photointermediates, which accumulate at steady state conditions.

****Minor Comments:****

p3, 1st line

"... of short and long wavelength light at safer light intensities."
Please specify what you mean with "safer light intensities".

p.4 lines 44-45

"Moreover, as these neurons are not readily silenced by a commonly applied channel-based optogenetic inhibitor (ArchT),..."
ArchT is not a channel-based optogenetic inhibitor. It rather is a light-driven, outwardly directed proton pump. Please correct.

p. 5 lines 11-14

"Second, in the majority of mCherry fluorescent cells tested, we found that this inhibition could be reversed by subsequent presentation of 525 nm light..."
Please give the precise number. What was the success rate?

p. 8 lines 22-23

"Lamplight activation in the inner retina supported rapid increases and decreases in firing of the retina's output neurons (RGCs)."
Please delete the word "rapid". This description is not supported by your results.

3. Significance:

Significance (Required)

Animal opsins allow the manipulation of intrinsic signaling pathways at comparatively low light intensities and therefore facilitate numerous optogenetic approaches. The authors used the recently discovered bistable Gi/o-coupled animal opsin Lamplight for neuronal inhibition in the hypothalamic suprachiasmatic nucleus and for photosensitization of degenerate retina. This is of general interest for the neuroscience community.

Field of expertise of co-reviewer:

Biophysics, Molecular Biology, Optogenetics, Neuroscience

Authors' Revision Plan

Reviewer #1 (Evidence, reproducibility and clarity (Required)):

****SUMMARY:****

Rodgers and Lucas and team demonstrate a new and powerful method for the optical regulation of neuronal activity in optogenetic experiments. This method consists of the heterologous expression of the parapinopsin protein from Lamprey, which is a bistable and predominantly Galpha-i/o-coupled opsin. This opsin is functionally characterized in a signaling assay in HEK293 cells and then for control of action potentials in SCN central neurons and retinal bipolar cell neurons. Key strengths of this opsin over other opsins are bistability (the ability to persist in the active state upon withdrawal of light, and the ability to terminate this state using a second colour of light) and an exceptional light sensitivity, and these are convincingly demonstrated in neuronal applications here.

****MINOR COMMENTS:****

I only have minor comments on control experiments and manuscript clarity.

Controls:

1)The authors apply three separate assays to demonstrate parapinopsin function in each with very good outcomes. Control experiments in which the examined cell types are exposed to light in the absence of the opsin appear to be missing. It would be desirable to add this data, which the authors certainly have already collected, to the manuscript.

We have added data from Hek293 cells not transfected with opsin to Fig 1C, 1G, 1H-J.

We have added data to supplementary figure 1, showing representative traces from SCN cells that do not express mCherry fluorescent reporter.

For the MEA data, we have included data from negative control retina as supplementary figure 2. Using same criteria as previously, we checked for light responses to 405nm stimulus from 82 spike-sorted units and identified 6 units with increases in firing to 405nm light. However, closer examination of these putative "light responsive units" revealed these either had a typical ipRGC response profile (slow onset, prolonged duration, only lasting first few trials) or were likely false positives. We did not find any units that matched the 3 response categories we describe in the Lamplight injected retinas.

2)Is addition of the rhodopsin-derived 1D4 epitope, which may contain phosphorylation sites, expected to impact parapinopsin function or expression?

We do not believe that the 1D4 epitope is having a substantial impact on the parapinopsin function or expression. The 1D4-tag is commonly used for expression of recombinant proteins in large volumes for purification¹. It is particularly useful for membrane proteins without commercially available high-affinity antibodies, including Lamprey parapinopsin²⁻⁴, which is why we have used it here. While the 1D4 tag does possess serines and threonines that could potentially be phosphorylated by native kinases in target cells, we do not believe this affects Lamplight function. If phosphorylation of the 1D4 tag was leading to deactivation

of the opsin by arrestin, we would expect to see termination of the Lamplight-driven response over time. Instead, in all three assays, we demonstrate Lamplight-driven activity in response to 405nm light can persist at a high level once light stimulus is removed and can be switched off in a controllable manner.

3)What is the maximal number of switching cycles ON/OFF that the authors have achieved using this opsin?

The maximum number of switching between ON and OFF we have recorded is 20 cycles. This was in the retina using multielectrode arrays. We have now added a representative unit to Figure 3 panel D and the following text to the manuscript on page 7 paragraph 1:

“Such switching behaviour could be repeated for at least the 20 stimulus repeats used in these recordings (Fig 3D)”.

Clarity:

1)On page 5 the authors state: "suggesting varying ratio of 405nm and 525nm light will lead to different degrees of hyperpolarization (Figure 2J)" - I am not convinced that Figure 2J is the dataset that supports this specific point.

We have made the following change to the text on page 6 paragraph 1 to clarify that we meant whether hyperpolarisation occurs is dependent on wavelength ratio:

“suggesting hyperpolarisation is dependent on the ratio of 405nm and 525nm light (Figure 2J)”

2)To assist with adaptation of this opsin in other laboratories, it could be important to clarify in which experiments exogenous 9 cis-retinal was added, and more generally in which types of experiments this will be recommended/required.

We used exogenous 9-*cis* retinal for the Hek293T BRET assay and retinal MEA experiments. It was not used in the SCN patching experiments. Based on the data collected from the SCN, we do not believe exogenous retinal is necessary for *in vivo* experiments

We have added the following text to page 5 paragraph 3 to clarify this:

“All SCN patching was conducted without addition of exogenous 9-*cis* retinal, suggesting it is not necessary for Lamplight-driven reversible hyperpolarisation in neurons.”

We have also added the following statement to our description of Lamplight’s qualities in the discussion on page 8 paragraph 1:

“SCN recordings were undertaken in normal aCSF (without additional retinoid) indicating that Lamplight functions, and can achieve high sensitivity, without addition of exogenous chromophore.”

3)On page 2, 2nd paragraph, middle: It may be pointed out that deactivation of other opsins is not only outside of the experimenter's control but often also rapid (which is undesirable if single-shot inhibition is required).

We have added this observation to the text on page 2, paragraph 2:

“This feature makes it hard to maintain a stable degree of opsin activation over time, and substantially limits experimental control over the timing of release from optogenetic

inhibition.”

4)On page 8, the authors state that this opsin combined with ratiometric stimulation may perform better than others when controlling a population of neurons with varied opsin expression; I am not sure if this is accurate: wouldn't ratiometric stimulation result in a greater or a smaller number of active opsins in cells that express higher or lower opsin levels, and thereby lead to different outcomes across the population?

We thank the reviewer for identifying this area of ambiguity in our manuscript. As the reviewer states, one source of variation is that different numbers of opsin molecules present in each cell will result in different numbers of G proteins activated. The referee is right that this source of variation cannot simply be overcome using ratiometric stimulation. Another important source of variation is the amount of light to which each cell is exposed (it is hard to evenly illuminate any 3D tissue). It is this latter source of error that the ratiometric control primarily addresses. It is also possible that non-linearities in the relationship between the amount of opsin activated and the integrated cellular response could allow the ratiometric approach also to reduce variation across cells with different levels of opsin expression in this scenario. However, as that is a rather complex (and currently theoretical) eventuality we feel it is easiest to simply rephrase this section on page 8 paragraph 3 as follows:

“Such ‘ratiometric’ control could be a useful complement to the standard method of controlling opsin activation (altering light intensity), as it has the potential to provide a more uniform level of activity across a population of neurons differing in the amount of light to which they are exposed.”

5)It would be desirable to express light intensities for some of the most important experiments also in wattage per area, which appears to be a common representation in the field.

We have added this information to the text in following locations:

In paragraph 3, page 3 (Hek293T cell experiments)

“Altering the intensity of the 405nm pulse modulated the BRET signal amplitude across a wide range of intensities, from 13-15.5 log photons/cm²/s, approximately 0.1μW/mm² to 30μW/mm² (Figure 1B). Lamplight sensitivity (log EC50 = 14.4log photons/cm²/s or 2.4μW/mm², Figure 1C) was broadly consistent with other animal opsins we have tested using BRET assay (Rod opsin log EC50 for 480nm = 14.6log photons or 3μW/mm²) [...] The threshold for a measurable response was 13.5 log photons/cm²/s, approximately 0.3μW/mm².”

In paragraph 2, page 5 (SCN patching experiments)

“We recorded this inhibitory response for a range of different 405nm light intensities from 14.5 to 12.9 log photons/cm²/s, equivalent to 3.8 to 0.11μW/mm²“

In paragraph 5, page 5

“We found 15.5 log photons/cm²/s, equivalent to 22μW/mm², of 525nm light”

In paragraph 4, page 6

“[...] we presented transduced retinas with sequential 2s flashes of 405nm (16.1log photons/cm²/s or 106uW/mm²) and 525nm (16.1log photons/cm²/s or 50uW/mm², 20s interstimulus interval).”

6) The authors cite the relevant literature in the field; perhaps one work of the Bruchas laboratory (Siuda Neuron 2015) and one of the Isacoff laboratory (Janovjak Nature Neuroscience 2009) could be additionally included and discussed, because these are relevant references for either the opiod-opsin chimeras mentioned on page 2 (Siuda) or for a bistable light-sensitive channel for neuronal inhibition (Janovjak).

We have added the suggested references and text discussing them to page 2 paragraph 1:

“HyLighter (Janovjak et al, 2009): a potassium-selective light-gated ion channel, also allows switchable inhibition that persists during dark, but requires high intensity ultraviolet light (380nm) and supplementation with azobenzene photoswitch MAG0 in order to function”

7) Minor improvements to the text could be considered, e.g. 10x => 10-fold (abstract), 405nm and 525nm => 405 and 525 nm, "may be used to switchable control" => "may be used to OBTAIN/ALLOW/ACHIEVE switchable control", "1s flashes of light" is repeated on page 7.

We have made the suggested improvements to text.

Reviewer #1 (Significance (Required)):

This study will have an exceptional impact on the field of optogenetics and neuroscience. Tools for reliable action potential silencing, in particular at low light levels, are not generally available. The data presented by the authors show clearly that this is possible with the lamprey parainopsin through a native-like G α -coupling mechanism. This study goes beyond this primary observation by demonstration of the additional benefit of bistability, and emerging from this, the ability tune activity levels by light composition, making this opsin a potential future go-to method in the field.

Reviewer #2 (Evidence, reproducibility and clarity (Required)):

The authors demonstrate the usability of the Gi/o-coupled, bistable animal opsin Lamplight for neuronal inhibition. Lamplight can be switched between signaling active and inactive states by the application of near UV- and green light pulses respectively, whereby the signaling active state triggers neuronal hyperpolarization via activation of the G-protein coupled inwardly rectifying potassium channel (GIRK). The authors show Lamplight-mediated, reversible suppression of spontaneous neuronal activity in the hypothalamic suprachiasmatic nucleus. Thereby Gi/o-mediated optogenetic silencing of neurons could be accomplished at comparatively low light intensities. Moreover, the authors show that Lamplight activation in mainly on-bipolar cells of rd1 mice, serving as a mouse model of retina degeneration, results in responses of opposite signs in retinal ganglion cells.

There are a few major concerns:

The authors seem to ignore relevant work of the Kleinlogel and Roska labs on optogenetic vision restoration. Moreover, statements on kinetics need to be balanced also in the abstract: "rapid" seems incompatible with sec time courses (see also below).

We have now added reference to Van Wyk et al (2015) and literature on use of microbial opsins for vision restoration, including Bi et al (2006), Lagali et al (2008), and Sengupta et al (2016), to the text.

We have changed the abstract text to read

“We conclude that Lamplight can co-opt endogenous signalling mechanisms to allow optogenetic inhibition that is scalable, sustained and reversible.”

p.2, second paragraph

This paragraph compares bistable animal opsins with non-bistable animal opsins. The optogenetic inhibition by Lamplight is described as "rapidly reversible" in the abstract. This statement should be corroborated in this paragraph by addressing the following question: How fast is the deactivation of Lamplight, when the deactivation process is accelerated by green light illumination, in comparison to the solely light-off switch triggered deactivation of other animal opsins like e.g. MW opsin (Berry et al., 2019, Eickelbeck et al., 2020)? Alternatively, optogenetic silencing should be described as "reversible" instead of "rapidly reversible" in the abstract.

The nature of deactivation is quite different in Lamplight (a direct light-induced change in structure) and animal opsins such as MW opsin (a two-step biochemical event - phosphorylation and arrestin binding, in addition to intrinsic deactivation by meta-II decay). Unfortunately, we do not yet have a direct measure of the structural changes in Lamplight or its kinetics (although limited experience with other bistable opsins, suggests a timescale in the ms range). The off kinetics of other opsins (including MW opsin) for optogenetic applications are likely to be quite variable (dependent on availability and local concentration of rhodopsin kinase and arrestins). The important issue we wished to raise in paragraph 2 is that timing of the former (light-switch) is under experimental control in a way that the latter can never be. Returning to the issue of speed of cellular response, as mentioned elsewhere, this is determined also by the kinetics of downstream elements of the G-protein signalling cascade. We used the qualitative description of ‘rapidly reversible’ in the abstract to highlight that a physiological change can be recorded with hundreds of ms (as shown by our retinal recordings). However, we do accept that the concept of ‘rapid’ is fundamentally relative and we therefore have altered the abstract as suggested.

p. 3 lines 38-40

"... this time course likely reflects the time taken for decay of the GRK3:beta-gamma complex, as light-induced changes in opsin conformation are near instantaneous. " The authors state that the time course of the BRET signal likely reflects the kinetics of the GRK3:beta-gamma complex formation/decay. Can the additional deactivation of highly light sensitive Lamplight molecules by BRET assay derived light sources, namely the light which is emitted by bioluminescent Nanoluc ($\lambda = 470 \text{ nm}$) and fluorescent venus ($\lambda = 535 \text{ nm}$), be ruled out? An additional deactivation due to the assay conditions would influence the time course of the BRET signal. Is the kinetics of the BRET signal independent of the recording times/intervals? If not, the potential influence of the assay conditions on the measured kinetics should be discussed.

Light from the BRET assay components NanoLuc and Venus is present continuously throughout the recording. Measurement of light emitted by the fluorescent Venus protein does not require additional light exposure, as the 470nm light emitted by Nanoluciferase acts as the excitation light for the fluorescent Venus, as part of the bioluminescence resonance energy transfer. Instead measurement of emitted light is acquired using photomultiplier tube

with suitable filters. Therefore, the recording intervals do not affect the light emitted by the assay components and will not change the kinetics of the response.

We cannot know for certain that these wavelengths (470 and 535nm) are not impacting deactivation of the opsin and it may be the case that this is contributing to the small decline in signal we see over time. However, we do not believe the Lamplight-driven response is substantially affected by the light emitted by the bioluminescent and fluorescent assay components, as we are able to measure a sustained and stable increase in G protein activation over tens of seconds (see top left panel from Fig1E below). In this example, Lamplight is activated by a 1s flash of 405nm light and cells are not exposed to any additional light after this point.

p. 5 lines 20-24

"Onset of hyperpolarization occurred rapidly to 405 nm with mean latency of 353.3{plus minus}92.6 ms and half-life of 273.9{plus minus}54.2 ms (Figure 2G-H, mean{plus minus}SEM). In comparison, it took several seconds for depolarisation by 525nm to begin, with mean latency of 3.5{plus minus}0.5s, and responses were much slower, with average half-life of 1.6{plus minus}0.3s (mean{plus minus}SEM)."

Please compare the kinetics of neuronal inhibition and disinhibition to the BRET signal derived kinetics which likely reflect the formation/decay of the GRK3:beta-gamma complex (Fig. 1). If known, please indicate which steps in the Gi/o pathway downstream of Lamplight activation and upstream of GIRK activation are rate limiting?

While we fully understand this referee's interest in relating the BRET assay to the neuronal inhibition, we argue that great care is needed here to avoid misleading the reader. The BRET signal in Hek293T cells, is produced by over-expressed G-protein subunits and GRK fragment. In a native cell, the relative concentration of these components will be different. Furthermore, the magnitude of change in activated G-protein required to produce a measurable change in BRET is likely substantially larger than required to alter cell physiology. For these reasons it is not possible to simply relate one method of recording to the other. From an application perspective the important data are those from the SCN and retinal recordings, which is what we concentrate on in the manuscript. We also would love to know which steps in the G-protein pathway are limiting in terms of kinetics, but addressing that problem is itself a substantial undertaking and we at present would prefer not to speculate. We have however, made several changes to the manuscript to improve our representation of this important issue:

We have amended the text in paragraph 4, page 3

“Note that, although this relaxation occurred over several seconds, this time course likely reflect assay-specific delays associated with the time taken for uncoupling of nanoLuc-GRK3 fragment from the beta-gamma dimer, as both previous (Eickelbeck et al., 2020) and our own experiments demonstrate faster kinetics are possible depending on assay and cell type used”

We have also added this point to the discussion in paragraph 2, page 8

“Light-induced transitions between active and inactive states for Lamplight (as all opsins) should be fast, but hysteresis in subsequent steps in the G-protein signalling cascade are expected to induce delays in the cellular response. Such delays will invariably be longer than for microbial opsins that more directly control ionic conductance. Nevertheless, neuronal responses to 405nm lamplight activating light can occur within a few hundred milliseconds in both SCN and retina. Deactivation following 525nm was more sluggish in the SCN, but was sufficiently fast in the retina to elicit a change in RGC firing within 500ms. The latter difference in response kinetics probably reflects fundamental properties of the G-protein signalling cascades engaged in the different cell types.”

p.6 lines 37-40

"The diversity of RGC responses to light in the Lamplight retina is consistent with the established ability of the retinal circuitry to invert polarity and apply different temporal frequency filters during signal transfer, and with the outcome of other studies restoring photosensitivity to ON bipolar cells. "

The demonstration of the usability of Lamplight for neuronal inhibition in the retina is impeded by the complexity of the system, in particular because the electrophysiological measurements were solely performed on RGCs. The impact of the paper would be substantially improved, if additional electrophysiological measurements were performed on Lamplight expressing bipolar cells while different light stimulation protocols are applied.

The proposed recordings are sadly not possible for us in the current climate. We do argue that they are not critical to the conclusions of our manuscript. We have extensively characterised the properties of Lamplight-driven activity in two different cell types with two different assays – G protein activation in Hek293T cells with the BRET assay and neuronal inhibition of SCN neurons with whole cell current clamp electrophysiology. In both these experiments we have used an array of different light stimulation protocols to explore how Lamplight driven-activity can be controlled.

Our goal with the retinal MEA experiments was to explore how Lamplight behaves within a more complex circuit, for an application in which close temporal control of opsin activity is especially important, and to inform whether it could have potential therapeutic applications for vision restoration. We decided to measure from retinal ganglion cells as the activity of these cells is most representative of the diverse visual information that would reach the brain in a treated individual. From these recordings, we were able to confirm that certain characteristic properties of Lamplight would be present in the retinal output– for example responses that can be scaled using ratiometric control and a proportion of sustained responses to 405nm that persist in dark until presentation of 525nm stimulus.

p.7 line 46 - p.8 lines 1-2

*"Such ratiometric control could be a useful complement to the standard method of controlling opsin activation by altering light intensity, as it has the potential to provide a more uniform level of activity across a population of neurons differing in the amount of opsin expressed, and lying at different distances from the light source."
Only UV-light activated Lamplight can be deactivated by additional green light illumination. The number of activated Lamplight molecules still depends on opsin expression and the distance from the light source. Why is the level of activity then considered being more uniform?*

We are grateful to the referee for identifying this lack of clarity. We have described our proposed solution in our response to Reviewer 1 above.

p. 8 lines 36-38

"Opsin photopigments have a stereotyped spectral sensitivity profile that is characterized by a much more complete loss of sensitivity at wavelengths longer vs shorter than the peak." This is true for groundstate activation. Please provide references, which show that this is also true for light activated photointermediates, which accumulate at steady state conditions.

The stereotyped spectral sensitivity profile we describe is based on the A1-visual pigment template⁵. This model is used to determine the lambda max of absorption spectra of both light activated and ground state opsins. For example, in the original paper describing Lamprey parapinopsin (Figure 2 of Koyanagi et al, 2008), for both the absorption spectra of the opsin in dark state and after exposure to UV light, relative absorbance is higher at wavelengths shorter than the peak, compared to those longer than the peak.

****Minor Comments:****

p3, 1st line

*"... of short and long wavelength light at safer light intensities."
Please specify what you mean with "safer light intensities".*

We have added additional text and references to clarify this statement:

"Here we show that these characteristics allow switchable and scalable inhibition of neurons using appropriate application of short and long wavelength light at safer light intensities than alternative optogenetic tools based on ion channels or pumps (Perny et al, 2016)"

p.4 lines 44-45

*"Moreover, as these neurons are not readily silenced by a commonly applied channel-based optogenetic inhibitor (ArchT),..."
ArchT is not a channel-based optogenetic inhibitor. It rather is a light-driven, outwardly directed proton pump. Please correct.*

We have corrected this.

p. 5 lines 11-14

"Second, in the majority of mCherry fluorescent cells tested, we found that this inhibition could be reversed by subsequent presentation of 525 nm light..."

Please give the precise number. What was the success rate?

We recorded from 5 mCherry fluorescent cell bodies. All showed hyperpolarisation by 405nm light with 2-14mV decrease in resting membrane potential. This range is likely down to variations in opsin expression inherent in the viral transduction technique. The 3 cells with a larger response (>5mV) also sustained inhibition, while the 2 cells with smaller responses recovered spontaneously before 525nm light was applied. All 3 cells with robust sustained inhibition could be reversed by 525nm light.

p. 8 lines 22-23

"Lamplight activation in the inner retina supported rapid increases and decreases in firing of the retina's output neurons (RGCs)."

Please delete the word "rapid". This description is not supported by your results.

We have replaced "rapid" with a more precise description of the kinetics:

"Lamplight activation in the inner retina supported increases and decreases in firing of the retina's output neurons within 500ms"

Reviewer #2 (Significance (Required)):

Animal opsins allow the manipulation of intrinsic signaling pathways at comparatively low light intensities and therefore facilitate numerous optogenetic approaches. The authors used the recently discovered bistable Gi/o-coupled animal opsin Lamplight for neuronal inhibition in the hypothalamic suprachiasmatic nucleus and for photosensitization of degenerate retina. This is of general interest for the neuroscience community.

Field of expertise of co-reviewer:

Biophysics, Molecular Biology, Optogenetics, Neuroscience

References

1. Molday, L. L. & Molday, R. S. 1D4 – A Versatile Epitope Tag for the Purification and Characterization of Expressed Membrane and Soluble Proteins. *Methods Mol. Biol. Clifton NJ* **1177**, 1–15 (2014).
2. Koyanagi, M. *et al.* Bistable UV pigment in the lamprey pineal. *Proc. Natl. Acad. Sci. U. S. A.* **101**, 6687–6691 (2004).
3. Kawano-Yamashita, E. *et al.* Activation of Transducin by Bistable Pigment Parapainopsin in the Pineal Organ of Lower Vertebrates. *PLoS One* **10**, e0141280 (2015).

4. Wada, S. *et al.* Color opponency with a single kind of bistable opsin in the zebrafish pineal organ. *Proc. Natl. Acad. Sci. U. S. A.* **115**, 11310–11315 (2018).
5. Govardovskii, V. I., Fyhrquist, N., Reuter, T., Kuzmin, D. G. & Donner, K. In search of the visual pigment template. *Vis. Neurosci.* **17**, 509–528 (2000).

Dear Jess,

Thank you for the submission of your revised manuscript. We have now received the comments from both referees and I am happy to say that both support its publication now. Only a few more minor editorial requests will need to be addressed before we can proceed with the official acceptance:

- Your manuscript has only 3 main figures and therefore needs to be published as a short report with a maximum of 29,000 characters (including spaces but excluding references and materials and methods) and combined results and discussion sections. Please combine these sections, which should also help to reduce the total character count.
- Please reduce the number of keywords to 5.
- Please add a "Data Availability Section" to the end of the materials and methods. If you have not deposited any data in public databases please mention this fact in this section.
- Please send us a completed author checklist that can be found here: <https://www.embopress.org/page/journal/14693178/authorguide> The completed author checklist will also be part of the transparent peer-review process file (RPF).
- The EMBO reports reference style has recently changed to Harvard style, please correct. Up to 10 authors need to be listed before "et al".
- Please upload all main and all EV figures as separate files. The supplementary figures need to be called Figure EV1, etc. Supplementary figures 2 and 3 are missing, please correct.
- The supplementary table should be uploaded as Table EV1. It may remain as a word file.
- There is a callout to Fig 3G, but there is no such panel.
- The Methods need to be called "Materials and Methods"
- The figure legends need to be moved to the end of the Article file.

I attach to this email a related manuscript file with comments by our data editors. Please address all comments in the final manuscript.

EMBO press papers are accompanied online by A) a short (1-2 sentences) summary of the findings and their significance, B) 2-3 bullet points highlighting key results and C) a synopsis image that is exactly 550 pixels wide and 200-600 pixels high (the height is variable). You can either show a model or key data in the synopsis image. Please note that text needs to be readable at the final size. Please send us this information along with the revised manuscript.

As part of the EMBO publication's Transparent Editorial Process, EMBO reports publishes online a

Review Process File (RPF) to accompany accepted manuscripts. This File will be published in conjunction with your paper and will include the referee reports, your point-by-point response and all pertinent correspondence relating to the manuscript.

I look forward to seeing a final version of your manuscript as soon as possible.
Please let me know if you have any questions or comments.

Kind regards,
Esther

Referee #1:

The authors have addressed all my comments in detail and satisfactorily.

The authors have also addressed the comments of the other reviewer in a manner that significantly improved the manuscript.

This manuscript is ready for publication.

Referee #2:

The points raised regarding evidence, reproducibility and clarity of the manuscript are adequately addressed. This study will certainly have an impact on the neuroscience community and therefore suits well for publication in EMBO reports.

The authors have addressed all minor editorial requests.

Dr. Jessica Rodgers
University of Manchester
Oxford Road
Greater Manchester M13 9PT
United Kingdom

Dear Dr. Rodgers,

I am very pleased to accept your manuscript for publication in the next available issue of EMBO reports. Thank you for your contribution to our journal.

At the end of this email I include important information about how to proceed. Please ensure that you take the time to read the information and complete and return the necessary forms to allow us to publish your manuscript as quickly as possible.

As part of the EMBO publication's Transparent Editorial Process, EMBO reports publishes online a Review Process File to accompany accepted manuscripts. As you are aware, this File will be published in conjunction with your paper and will include the referee reports, your point-by-point response and all pertinent correspondence relating to the manuscript.

If you do NOT want this File to be published, please inform the editorial office within 2 days, if you have not done so already, otherwise the File will be published by default [contact: emboreports@embo.org]. If you do opt out, the Review Process File link will point to the following statement: "No Review Process File is available with this article, as the authors have chosen not to make the review process public in this case."

Should you be planning a Press Release on your article, please get in contact with emboreports@wiley.com as early as possible, in order to coordinate publication and release dates.

Thank you again for your contribution to EMBO reports and congratulations on a successful publication. Please consider us again in the future for your most exciting work.

THINGS TO DO NOW:

You will receive proofs by e-mail approximately 2-3 weeks after all relevant files have been sent to our Production Office; you should return your corrections within 2 days of receiving the proofs.

Please inform us if there is likely to be any difficulty in reaching you at the above address at that time. Failure to meet our deadlines may result in a delay of publication, or publication without your corrections.

All further communications concerning your paper should quote reference number EMBOR-2020-51866V2 and be addressed to emboreports@wiley.com.

Should you be planning a Press Release on your article, please get in contact with emboreports@wiley.com as early as possible, in order to coordinate publication and release dates.

YOU MUST COMPLETE ALL CELLS WITH A PINK BACKGROUND ↓
PLEASE NOTE THAT THIS CHECKLIST WILL BE PUBLISHED ALONGSIDE YOUR PAPER

Corresponding Author Name: Robert Lucas

Manuscript Number: EMBOR-2020-51866V1